# Intestinal *Klebsiella pneumoniae* Contributes to Pneumonia by Synthesizing Glutamine in Multiple Myeloma

**DOI:** 10.3390/cancers14174188

**Published:** 2022-08-29

**Authors:** Yihui Wang, Qin Yang, Yinghong Zhu, Xingxing Jian, Jiaojiao Guo, Jingyu Zhang, Chunmei Kuang, Xiangling Feng, Gang An, Lugui Qiu, Guancheng Li, Yanjuan He, Wen Zhou

**Affiliations:** 1State Key Laboratory of Experimental Hematology, National Clinical Research Center for Geriatric Disorders, Haihe Laboratory of Cell Ecosystem, Department of Hematology, Xiangya Hospital, Central South University, Changsha 410008, China; 2Key Laboratory for Carcinogenesis and Invasion, Chinese Ministry of Education, Key Laboratory of Carcinogenesis, Chinese Ministry of Health, Cancer Research Institute, School of Basic Medical Sciences, Central South University, Changsha 410008, China; 3Department of Hematology, The Third Xiangya Hospital, Central South University, Changsha 410013, China; 4Xiangya School of Public Health, Central South University, Changsha 410028, China; 5State Key Laboratory of Experimental Hematology, Institute of Hematology & Blood Diseases Hospital, Chinese Academy of Medical Science & Peking Union Medical College, Tianjin 300020, China

**Keywords:** multiple myeloma, pneumonia, gut microbiome, *Klebsiella pneumonia*, glutamine

## Abstract

**Simple Summary:**

Multiple myeloma (MM) is characterized by the presence of systemic clinical manifestations. Among them, pneumonia accounts for a significant cause of morbidity and mortality, highlighting an urgent need to explore possible susceptibility and potential mechanisms of pneumonia in MM. Recently, the gut–lung axis has emphasized the relevance of gut microbiota and pneumonia, indicating that the compromised function of gut microbiota may be susceptible to pneumonia. It has been previously shown that intestinal *Klebsiella pneumonia* enriches in MM patients and promotes MM progression in our previous work. In addition, host metabolism has been identified as a key regulator of MM. However, what role the altered gut microbiota plays in MM with pneumonia remains unknown. In this study, we explored the association between gut microbiota and MM with pneumonia. This is the first study to identify and elucidate the function of gut microbiota involved in MM with pneumonia.

**Abstract:**

Pneumonia accounts for a significant cause of morbidity and mortality in multiple myeloma (MM) patients. It has been previously shown that intestinal *Klebsiella pneumonia* (*K. pneumonia*) enriches in MM and promotes MM progression. However, what role the altered gut microbiota plays in MM with pneumonia remains unknown. Here, we show that intestinal *K. pneumonia* is significantly enriched in MM with pneumonia. This enriched intestinal *K. pneumonia* links to the incidence of pneumonia in MM, and intestinal colonization of *K. pneumonia* contributes to pneumonia in a 5TGM1 MM mice model. Further targeted metabolomic assays reveal the elevated level of glutamine, which is consistently increased with the enrichment of *K. pneumonia* in MM mice and patients, is synthesized by *K. pneumonia*, and leads to the elevated secretion of TNF-α in the lung normal fibroblast cells for the higher incidence of pneumonia. Inhibiting glutamine synthesis by establishing *glnA*-mutated *K. pneumonia* alleviates the incidence of pneumonia in the 5TGM1 MM mice model. Overall, our work proposes that intestinal *K. pneumonia* indirectly contributes to pneumonia in MM by synthesizing glutamine. Altogether, we unveil a gut–lung axis in MM with pneumonia and establish a novel mechanism and a possible intervention strategy for MM with pneumonia.

## 1. Introduction

Multiple myeloma (MM) is a malignancy of terminally differentiated plasma cells and is characterized by the presence of systemic clinical manifestations [1]. Recent findings have shown that newly diagnosed MM (NDMM) patients have an increased susceptibility to infection [2,3]. Seriously, it has been reported that approximately 10% of NDMM patients die prematurely from infection before they have had access to receive effective therapies [4,5,6]. Therefore, infection is regarded as a significant cause of morbidity and mortality in MM patients [7]. As reported in multiple investigations, pneumonia is the most frequent bacterial infectious complication [2,8,9], which results in increased hospitalization time, worse outcomes, and devastating consequences. Hence, there is an exigent need to explore susceptible factors and potential mechanisms of pneumonia in MM.

Recently, evidence has highlighted the relevance of gut microbiota and pneumonia, referred to as the gut–lung axis [10,11,12]. Descriptively, gut microbiota reportedly defends against pneumonia by the effects of microbiota on host immunity and the role of microbial metabolite on the lung [13,14]. The conventional views hold that the intestine plays a detrimental role by facilitating systemic inflammation and infection in critically ill conditions [15]. Nevertheless, particular disease states and antibiotic therapies may unbalance the gut microbiota, therefore emerging different conclusions. In short, research indicates that the compromised function of gut microbiota may be susceptible to pneumonia in some cases.

In our previous work, it was estimated that intestinal nitrogen-recycling bacteria were significantly enriched in MM, accelerating MM progression [16]. In the same study, we observed that enriched *K. pneumonia* promoted MM progression via de novo synthesis of glutamine in vivo. In addition, in another study of our recent work, glycine was identified as a key metabolic regulator of MM, unveiling the role of host metabolism in MM [17]. These discoveries in the impact of the gut microbiome and host metabolism on MM can thereby have considerable implications for the possible association between pneumonia in MM and gut microbiota. What role the altered gut microbiota play in pneumonia, however, remains unknown.

In this study, we attempt to fill the gap in our knowledge of gut microbiota involved in MM with pneumonia. We screened the differential bacteria by metagenomic sequencing, followed by examining alterations of microbial metabolite for pneumonia in MM.

## 2. Materials and Methods

### 2.1. Patient Samples

Human feces and serum samples were obtained from NDMM patients in the Xiangya and Third Xiangya Hospital, Central South University (Changsha, China), and the Institute of Hematology and Blood Disease Hospital, Chinese Academy of Medical Sciences and Peking Union Medical College (Tianjin, China). The procedure was informed and consented to by all participants and approved by the Cancer Research Institute of the Central South University Medical Ethics Committee. Collected fresh fecal and serum samples were stored at −80 °C for further detection.

Based on clinical symptoms, chest radiography, and etiological detection, the enrolled NDMM patients without any other disease or infection and any clinical treatment, including treatment of antibiotics, were classified into without or with pneumonia, which was confirmed by the experienced clinicians. Among them, fresh fecal samples from 5 MM patients without pneumonia and 4 MM patients with pneumonia were sampled for metagenomic sequencing, while serum from 115 MM patients (without/with pneumonia: 73/42) was sampled for targeted metabolomics analysis. What is more, 29 MM patients (without/with pneumonia: 15/14) were defined as an expanded cohort for further validation of metagenomic sequencing results. In addition, paired fecal and serum samples in another 15 NDMM patients (without/with pneumonia: 7/8) were collected to define the association between intestinal *K. pneumonia* and glutamine.

### 2.2. Cell Lines, Antibodies, and Reagents

The 5TGM1 MM cells were cultured in RPMI 1640 medium (Gibco, MA, USA) supplemented with 10% fetal bovine serum (Biological Industries, Kibbutz Beit-Haemek, Israel). The human lung normal fibroblast cells, WI38 cells, provided by Dr. Yu Sun (Shanghai Institute of Nutrition and Health, Chinese Academy of Sciences, China), were cultured in high-glucose DMEM (Gibco, MA, USA) supplemented with 10% fetal bovine serum.

Anti-GAPDH (#10494-1-AP) and anti-TNF-α antibodies (#60291-1-Ig) were from Proteintech Group (Rosemont, IL, USA). Anti-α-SMA antibody (#ab7817) was obtained from Abcam (Cambridge, UK). Horseradish peroxidase (HRP)-conjugated secondary goat anti-rabbit (#L3012) and goat anti-mouse (#L3032) antibodies were purchased from Signalway Antibody (College Park, MD, USA). 

### 2.3. Metagenomic Sequencing and Identification of Differential Species

The total DNA from individual fecal samples was extracted utilizing the E.Z.N.A Stool DNA extraction kit (Omega Bio-Tek, Norcross, GA, USA) under the manufacturer’s instructions. Then, metagenomic sequencing was performed to screen differential species utilizing the R platform. The significant difference was defined as the *p*-value of the Wilcoxon rank-sum test < 0.05.

### 2.4. The 16S Ribosomal DNA qPCR

Referring to our previous experience and published paper [16,18], the paired primers specific for each species and conserved ones for total bacteria were designed and are presented in Appendix A. Total bacterial DNAs were extracted from fecal samples and qPCR implemented to confirm the relative abundance of differential species following the previously published method [16].

### 2.5. The 5TGM1 MM Mouse Model

All animal experiments were designed and conducted to conform to the guidelines of the Institutional Animal Care and local veterinary office and ethics committee of Central South University, China (animal experimental license: CSU-2022-0256) under an approved protocol. The luciferase-expressing 5TGM1 MM cells (1 × 10^6^ cells in 200 μL PBS per mouse) were injected into 6-week-old C57BL/KaLwRij mice to build a 5TGM1 MM mouse model (dubbed “5TGM1 MM mice”) as reported before [16,17]. For Figure 1, female mice were adopted, while for the other in vivo experiment, male mice were utilized. MM progression was monitored by measuring serum IgG2b concentration via ELISA (Bethyl Laboratories, Inc., Montgomery, TX, USA). The clinical endpoint was defined as the exhibition of hind limb weakness in partial mice.

### 2.6. Construction and Confirmation of K. pneumonia Containing Mutant glnA

The mutant *K. pneumonia* (*glnA*, coding glutamine synthetase) was constructed by homologous recombination, as previously published [16]. Amplification by PCR and then detection by DNA agarose gel electrophoresis, as well as an examination of the expression of *K. pneumonia*-*glnA*, were performed to confirm the successful construction.

### 2.7. Intestinal Colonization with K. pneumonia In Vivo 

All C57BL/KaLwRij mice were treated with an antibiotic cocktail (ABX) in the drinking water for one week, as previously reported [19]. Then, mice were given oral gavage of 2 × 10^8^
*K. pneumonia* (BNCC 102997 = ATCC 10031) or vehicle (PBS). The *K. pneumonia* was purchased from Beijing BeiNa Biotechnology Institute and cultured in nutrient broth (Product ID: 022010, Guangdong HuanKai Microbial Co., Ltd., Guangzhou, China) at 37 °C and 220 rpm for 16 h. After 1 week of gavage, mice were challenged with 5TGM1-Luc cells. After that, the gavage was conducted every 2 days until the endpoint to maintain the colonized state. Feces and serum were collected every two weeks. The 16S ribosomal DNA qPCR was performed to confirm the successful intestinal colonization of *K. pneumonia*. At the experimental endpoints, mice were sacrificed for cecal contents and lung specimen harvest. For the assessment of lung inflammation, inflammatory infiltration by H&E staining, total lung histopathology scores, and expressions of typical inflammatory factors in lung tissue were detected.

### 2.8. Immunohistochemistry

Immunohistochemistry was performed on 3 μm sections obtained from formalin-fixed paraffin-embedded lung tissues. The process was carried out as previously described [16]. The tissue sections were incubated with primary antibodies of TNF-α (1:2000 final dilution) and α-SMA (1:500 final dilution). The stained sections were evaluated using PerkinElmer Quantitative Pathology Imaging System and the inForm software (PerkinElmer, Inc., Waltham, MA, USA) for segmentation and quantification of TNF-α+/α-SMA+ cells.

### 2.9. Targeted Metabolomic Assays 

Targeted metabolomics assays of peripheral blood (PB) plasma were performed by using the Q300 Metabolite Assay Kit (Human Metabolomics Institute, Inc. Shenzhen, China), as previously published [17,20]. The resulting datasets were analyzed by the SIMCA-P 14.1 software (Sartorius Stedim Biotech, Germany) package and MetaboAnalyst 5.0 online software (https://www.metaboanalyst.ca/). Orthogonal partial least-squares-discriminant analysis (OPLS-DA) was performed to visualize the metabolic alteration. Then, variable importance of the projection (VIP) values from the OPLS-DA model and the fold changes (FC) and *p*-values of metabolites with MetaboAnalyst 5.0 were obtained, respectively. Differential metabolites (*p* < 0.05) were defined. In addition, measurements of differential metabolites were performed utilizing high-performance liquid chromatography (HPLC), as described previously [17,21]. 

### 2.10. Real-Time Quantitative PCR

Total RNA was extracted from whole lung cells and WI38 cells using Trizol. cDNA was synthesized from 1 μg total RNA and real-time qPCR for TNF-α, IL-1β, and IL-6 was performed as previously presented [22]. The primer sequences were provided in Appendix A.

### 2.11. Plate Colony and CCK8 Assay

The colony-formation ability and proliferation ability of WI38 cells intervened with glutamine were compared in terms of plate colony and CCK8 assay. WI38 cells were planked with DMEM complete medium in the different concentrations of glutamine. For plate colony and CCK8 assay, conventional procedures were employed. Plates were imaged and colonies were enumerated by using Image J 1.46r software (National Institutes of Health, USA).

### 2.12. Western Blotting

Western blot analysis was performed as described previously [22,23]. Proteins were separated on 12% sodium dodecyl sulfate-polyacrylamide gel electrophoresis and then transferred to a polyvinylidene fluoride membrane. The membrane was stained with the corresponding primary and secondary antibodies. The specific protein bands were detected using SuperSignal West Pico Chemiluminescent Substrate (Pierce, Rockford, IL, USA).

### 2.13. Statistical Analysis

Quantitative data are shown as means ± SEM. Comparisons between groups were analyzed by Student’s *t*-test, one-way ANOVA test, and Chi-square analysis (GraphPad Prism 8). Paired Student’s *t*-test was employed to define differences in intestinal colonization of *K. pneumonia*. Pearson correlation analysis was utilized to test the correlation between the two groups. Statistically significant differences are indicated as follows: * *p* < 0.05, ** *p* < 0.01, *** *p* < 0.001, **** *p* < 0.0001.

## 3. Results

### 3.1. Subsection K. pneumonia Is Enriched in the Intestine and Is Linked to MM with Pneumonia

To figure out the incidence of pneumonia in NDMM, we retrospectively summarized all NDMM cases at Xiangya Hospital in the past year. The representative chest radiography was exhibited and the incidence of pneumonia in NDMM patients was 44% (Figure 1a). Then, shotgun metagenomic sequencing of fecal samples was performed and 10 differential bacteria were identified (Figure 1b), all of which were intersected by Venn diagram analysis with bacteria differentially enriched in NDMM patients compared with healthy donors [16], leading to the coincident differential bacteria *K. pneumonia* (Figure 1c). Subsequently, the higher intestinal abundance of *K. pneumonia* (abbreviated as *KPn*) in NDMM patients with pneumonia was verified in an expanded cohort (Figure 1d).

Subsequently, to confirm the above-mentioned result in vivo, a 5TGM1 MM mice model was established and C57BL/KaLwRij mice were injected intravenously with PBS as a comparison (Figure 1e). To reflect the tumor burden, we monitored serum IgG2b concentration, which was secreted by 5TGM1 myeloma cells. The concentrations of serum IgG2b in the MM mice were much higher than in the controlled group (Figure 1f), indicating the successful establishment of the mouse model. At the endpoint, H&E staining and expressions of typical inflammatory factors of the lung were performed to examine the inflammation. As a consequence, more inflammatory infiltrations (Figure 1g,h), accompanied by higher expressions of TNF-α, IL-1β, and IL-6 were demonstrated in MM mice (Figure 1i–k). As expected, intestinal *K. pneumonia* was enriched in MM mice (Figure 1l). Moreover, we found that intestinal *K. pneumonia* abundance correlated strongly with the incidence of pneumonia in our enrolled cohort (*p* = 0.0418), while no differences existed in other patients’ characteristics (e.g., the percentage of plasma cells, tumor stages, etc.) (Table 1). In short, these findings suggested that enrichment of intestinal *K. pneumonia* is linked to pneumonia in MM in vivo.

### 3.2. Intestinal K. pneumonia Accelerates Pneumonia in 5TGM1 MM Mice

To further testify whether intestinal *K. pneumonia* contributed to pneumonia in MM, continuous intestinal colonization of *K. pneumonia* was induced in 5TGM1 MM mice (Figure 2a). The successful intestinal colonization of *K. pneumonia* was confirmed at week 0 (Figure 2b). Moreover, the higher concentration of serum IgG2b was tested in *KPn* mice (Figure 2c), demonstrating that *K. pneumonia* prompted MM tumor burden. Furthermore, more obvious inflammatory infiltration by H&E staining (Figure 2d,e), higher expression of TNF-α and α-SMA by immunohistochemistry (Figure 2f), and significantly increased expressions of TNF-α, IL-1β, and IL-6 by qPCR (Figure 2g–i) were exhibited in *KPn* mice compared with controls, demonstrating more inflammation in the lung tissue in the *KPn* mice. The above-mentioned results described the more serious pneumonia in *KPn* mice, hinting that intestinal *K. pneumonia* accelerated pneumonia in 5TGM1 MM mice.

### 3.3. Glutamine Is Elevated in PB Plasma of MM Patients with Pneumonia

We next seek to understand what *K. pneumonia* does in MM with pneumonia. Targeted metabolomic assays were performed to study the differential metabolites in PB plasma derived from NDMM patients with or without pneumonia. OPLS-DA models were established and showed differences in the two groups (Figure 3a). Then, differential metabolites (*p* < 0.05) were explored (Figure 3b), whose detailed VIP values, FCs, and *p*-values are shown in Appendix A. Among them, it has been reported that *K. pneumonia* de novo synthesizes glutamine by utilizing NH_4_^+^ or urea [16]. Thus, serum glutamine concentrations of NDMM patients were examined, presenting higher levels of serum glutamine in pneumonia subjects (Figure 3c). Specifically, the characteristics of these subjects were analyzed, suggesting glutamine concentrations correlated strongly with the incidence of pneumonia (*p* = 0.0093) (Table 2). Further, to define the association between intestinal *K. pneumonia* and glutamine, paired fecal and PB plasma samples in NDMM patients were collected. It appeared that the intestinal abundance of *K. pneumonia* was positively correlated with PB plasma glutamine concentrations (Figure 3d). Then, the mutant *glnA* containing *K. pneumonia* was constructed to reduce the synthesis of glutamine (Figure 3e). The successful construction of mutated *K. pneumonia* was confirmed by the DNA agarose gel electrophoresis of *glnA* (Figure 3f), and relative expression of *K. pneumonia*-*glnA* (Figure 3g). Subsequently, an in vivo experiment with the mutated strain was carried out.

### 3.4. Intestinal K. pneumonia Synthesizes Glutamine to Bolster Pneumonia in MM In Vivo

To further explore whether the glutamine synthesized by intestinal *K. pneumonia* promoted pneumonia in MM in vivo, we performed bacterial transplantation by setting up three groups: PBS, colonization of *K. pneumonia* with wild type (*KPn*^WT^), and mutated *glnA* (*KPn*^Mut^) (Figure 4a). At week 0, successful colonization of *K. pneumonia* was confirmed (Figure 4b). Then, a higher concentration of IgG2b in *KPn*^WT^ mice than that in *KPn*^Mut^ and PBS mice was detected (Figure 4c). Next, more obvious inflammatory infiltration by H&E staining (Figure 4d,e), higher expression of TNF-α and ɑ-SMA by immunohistochemistry (Figure 4f), and significantly increased expressions of TNF-α and IL-6 by qPCR (Figure 4g,h) were shown in *KPn*^WT^ mice than in *KPn*^Mut^ mice, indicating less serious inflammation in the glutamine-reduced group. Next, lower levels of glutamine were detected in the cecum and lung in *KPn*^Mut^ mice (Figure 4i,j), and the cecum glutamine was positively related to the lung glutamine (Figure 4k). We thus consider that glutamine synthesized by intestinal *K. pneumonia* promoted pneumonia in MM.

### 3.5. Glutamine Contributes to Pneumonia by Promoting TNF-α Expression of WI38 Cells

Subsequently, to question the effects of glutamine on WI38 cells, additional glutamine was supplemented. The results showed glutamine facilitated WI38 cell proliferation by plate clone (Figure 5a,b), as well as via CCK8 detection (Figure 5c). To validate whether glutamine was associated with inflammation, expressions of TNF-α, IL-1β, and IL-6 of WI38 cells with an additional glutamine supplement were detected. As a result, significantly increased expression was shown in TNF-α (Figure 5d,e). The results indicated a positive role of glutamine on TNF-α expression of WI38 cells, which was then supported by the CCK8 detection of the addition of anti-TNF-α, showing the glutamine indeed promoted TNF-α activation (Figure 5f). The results further testified that glutamine concentration in the lung was positively correlated with the lung TNF-α mRNA expression in vivo (Figure 5g). These available results suggested glutamine produced by *K. pneumonia* prompted TNF-α expression of WI38 cells, thereby accelerating pneumonia (Figure 6).

## 4. Discussion

Our results conclude that intestinal *K. pneumonia* plays a detrimental role in pneumonia in MM, as evidenced by elevated bacterial abundance in NDMM patients with pneumonia, and serious pneumonia in MM mice colonized by *K. pneumonia*. Further targeted metabolomic assays reveal the elevated level of glutamine, which is synthesized by intestinal *K. pneumonia* to indirectly exert effects on MM, and leads to the elevated secretion of TNF-α in the lung, thereby promoting pneumonia in MM. In line with our previous publication, wherein the intestinal *K. pneumonia* was found to accelerate MM progression [16], this work particularly focuses on the most common clinical manifestation of infection in MM. To our knowledge, a gut–lung axis in MM with pneumonia is firstly underscored by this work, which establishes a novel mechanism and a possible intervention strategy for pneumonia in MM. 

As previously introduced, MM patients have an increased susceptibility to pneumonia. In our retrospective analysis, the incidence of pneumonia in NDMM is 44%. Studying pneumonia in MM has been challenging, but the recent identification of a gut–lung axis during pneumonia brings us new insight to understand pneumonia. Emerging evidence indicates the role of intestinal microbiota in its responses to respiratory infections [12,24,25]. As reported, Bifidobacterium spp. protected mice from pulmonary infection [26,27]. In this line, probiotics improved the incidence and outcomes of respiratory infections in humans [28,29,30,31]. Unlike the protection role of commensal microbiota on pneumonia, we uncovered for the first time that the pathological bacteria *K. pneumonia* is enriched in the intestine of MM patients with pneumonia and plays a negative role. It remains to be established, however, whether the protection of decreasing commensal microbiota is responsible for this effect. 

Several studies have demonstrated multiple means of communication between the gut and the lung. Notably, Clarke et al. were among the first to propose that lung innate immune responses can be reinforced by the translocation of intestinal microbiota-derived products [32]. Further, evidence is increasing in support that the effects of gut commensal bacteria on lung innate and adaptive immunity to protect from pneumonia [10,33,34,35]. Furthermore, current research has uncovered that SCFAs have been shown to exert a positive effect against respiratory tract infections, mainly by anti-inflammatory properties [13,36,37]. In our study, we first established the model of glutamine synthesized by intestinal *K. pneumonia* based on patients’ specimens and *glnA* mutant *K. pneumonia in vivo* experiments. This enabled us to demonstrate that the microbial metabolite glutamine, which is closely correlated with an altered *K. pneumonia* abundance, has a significant influence on lung infection in MM. To our knowledge, MM cells show features of glutamine addiction and depend on extracellular glutamine for proliferation and progression [16,38,39]. The function of enriched glutamine in MM lung inflammation is yet not to be defined. In our study, we further extend the function of glutamine in MM by promoting pneumonia, as reflected by enhanced inflammatory responsiveness and an increased secretion of TNF-α. Nevertheless, we did not investigate whether glutamine exerts effects on immune cells, which was regarded as the main mechanism of the intestine and lung communications. In addition, it remains to be determined whether the observed effects are caused by translocation of intestinal *K. pneumonia*. We failed to testify this possibility because of the technical limitations of the bacteria tracing experiment (data not shown). More experiments are required to confirm it.

The role of diet in determining gut microbiota has been studied [33]. As previously reported by our work, the elevation of glycine in the bone marrow, which is due to bone collagen degradation and exogenous dietary intake, contributes to MM progression [17]. In turn, NH_4_^+^, produced by nitrogen resources, was accumulated in the intestine, and then altered the intestinal bacterial composition, resulting in the preferential proliferation of nitro-recycling bacteria (*K. pneumonia*). Meanwhile, enriched nitro-recycling bacteria were verified to promote MM progression [16]. Moreover, glutamine synthesized by *K. pneumonia* leads to the elevated secretion of TNF-α in the lung normal fibroblast cells for a higher incidence of pneumonia. Altogether, diet, especially nitrogen-dietary intake, may promote intestinal nitro-recycling bacteria enrichment, which in turn forms a vicious cycle with MM cells that promotes MM progression. 

The finding of our study has led to the hypothesis that pneumonia might be dampened in MM, likely by balancing the gut microbiota or intervening microbial metabolites, e.g., reducing the abundance of intestinal *K. pneumonia* by supplementing probiotics, and intervening glutamine by limitation in dietary intake or inhibiting synthesis. In our study, it remains to be explored whether probiotics supplementing or reducing glutamine intake are responsible for the remission of pneumonia. In the future, these strategies would be further illustrated in clinical trials. 

The present study had the following strengths: metagenomic sequencing and targeted metabolomic assays were performed to screen differential bacteria and metabolites. Moreover, NDMM patients from multiple centers were enrolled to explore the association between intestinal *K. pneumonia* and glutamine, and together with *K.-pneumonia*-containing mutant *glnA* was established to further confirm our assumptions. Furthermore, we firstly identify a gut–lung axis in MM with pneumonia, which establishes a novel mechanism and a possible intervention strategy for pneumonia in MM. Nevertheless, our research had the following limitations. Firstly, the size of samples for metagenomic sequencing analysis was small, and further enlarged investigation with multiple centers will be undertaken. Secondly, the direct relationship between intestinal *K. pneumonia* and pneumonia development in MM is not really established in this study, such as that we failed to discover whether intestinal *K. pneumonia* was translocating from gut to lung. More evidence is required to further confirm it. Thirdly, an intervening strategy in clinical research should be carried out through balancing the gut microbiota by administration with probiotic supplementation and engineering bacteria transformation in combination with the first-line treatment in MM in the future.

## 5. Conclusions

In summary, our present findings demonstrate that enriched intestinal *K. pneumonia* is associated with the incidence of pneumonia in MM and contributes to pneumonia by synthesizing glutamine to promote TNF-α expression. Taken together, the present study highlights the gut–lung axis in MM with pneumonia and reveals a novel mechanism and a potential therapeutic strategy for MM patients with pneumonia.

## Figures and Tables

**Figure 1 cancers-14-04188-f001:**
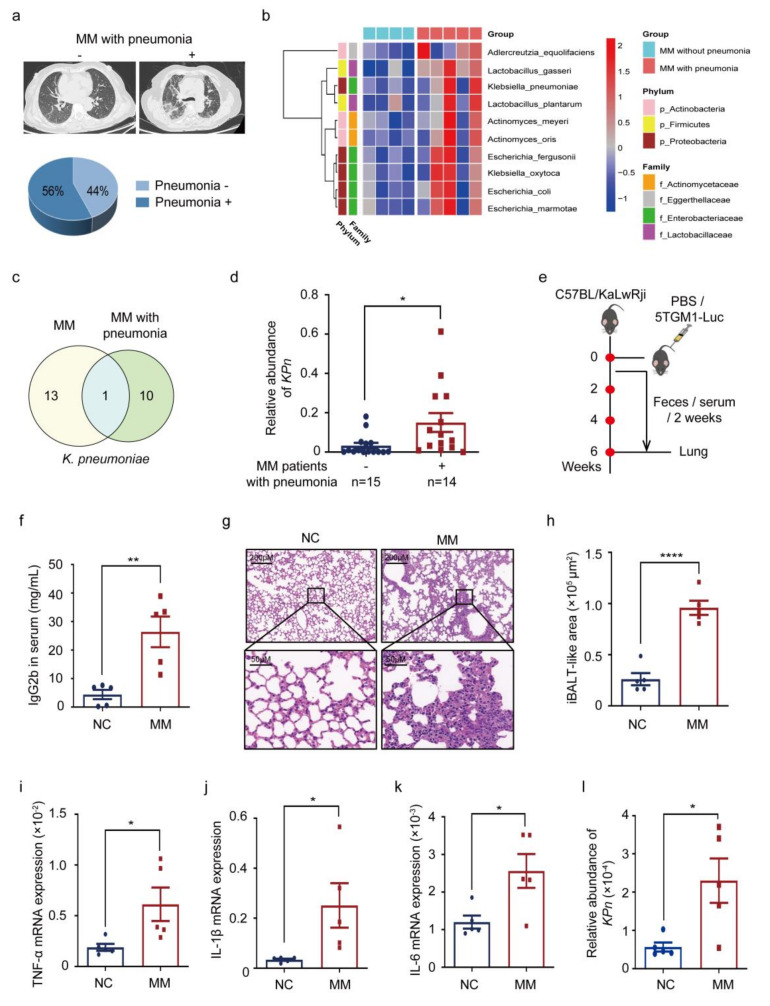
*K. pneumonia* is enriched in the intestine and is linked to MM with pneumonia. (**a**) Representative images of chest computed tomography from MM patients without pneumonia and with pneumonia, and incidence of pneumonia in newly diagnosed MM patients in Xiangya Hospital in the past year; (**b**) heatmap of intestinal differential bacteria in newly diagnosed MM patients without pneumonia (n = 4) and with pneumonia (n = 5) by shotgun metagenomic sequencing of fecal samples; (**c**) Venn diagram of coincident differential bacteria between healthy donors vs. newly diagnosed MM and newly diagnosed MM without vs. with pneumonia; (**d**) relative abundance of intestinal *K. pneumonia* in MM patients without/with pneumonia in an expanded MM cohort (mean ± SEM); (**e**) schematic of in vivo experimental workflow; (**f**) the concentrations of serum IgG2b in control mice (NC) and 5TGM1 MM mice (MM) by ELISA (mean ± SEM); (**g**) H&E staining of the lung from NC and MM group; (**h**) the area of iBALT-like (Inducible bronchus-associated lymphoid tissues) area in NC and MM group (mean ± SEM); (**i**–**k**) mRNA expression of TNF-α, IL-1β, and IL-6 in lung from NC and MM group by using qPCR (mean ± SEM); (**l**) relative abundance of intestinal *K. pneumonia* in NC and MM mice by qPCR (mean ± SEM); * *p* < 0.05, ** *p* < 0.01, **** *p* < 0.0001; *p*-values were calculated using two-tailed unpaired Student’s *t*-tests.

**Figure 2 cancers-14-04188-f002:**
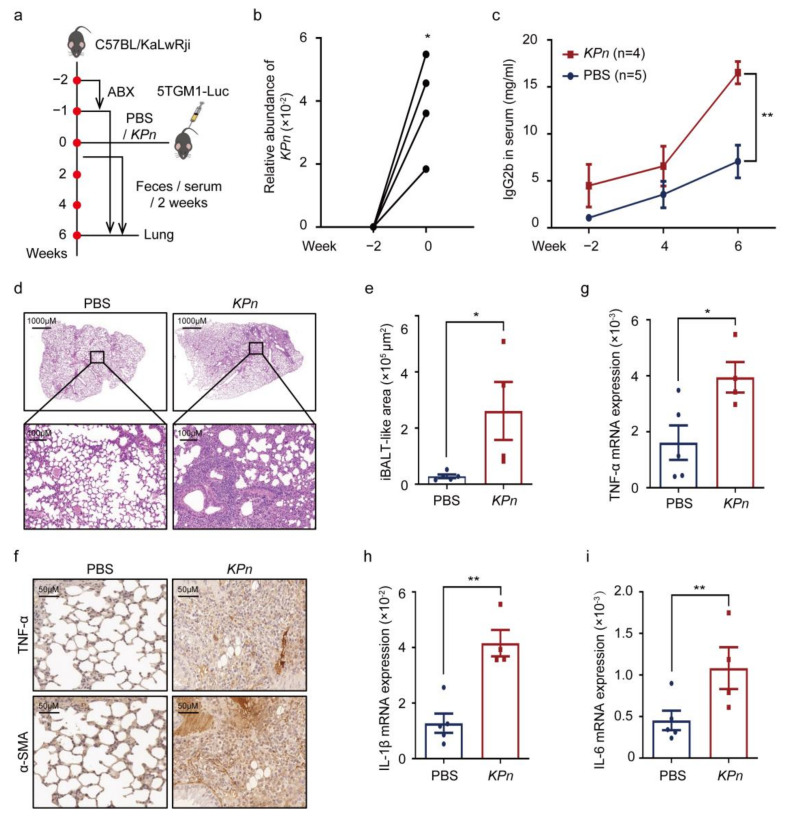
Intestinal *K. pneumonia* accelerates pneumonia in 5TGM1 MM mice. (**a**) Schematic of intestinal *K. pneumonia* transplantation experimental workflow; (**b**) relative abundance of intestinal *K. pneumonia* in *K. pneumonia* colonized mice at week −2 and week 0; (**c**) serum IgG2b concentrations in PBS gavaged mice (PBS) and *K. pneumonia* colonized mice (*KPn*) by ELISA (mean ± SEM); (**d**) representative images of H&E staining of the lung from PBS and *KPn* group; (**e**) the iBALT-like area in PBS and *KPn* group (mean ± SEM); (**f**) representative images of immunohistochemistry staining of TNF-α and α-SMA in the lung from PBS and *KPn* group; (**g**–**i**) mRNA expression of TNF-α, IL-1β, and IL-6 in the lung from PBS and *KPn* group by using qPCR (mean ± SEM); * *p* < 0.05, ** *p* < 0.01; *p*-values were calculated using two-tailed unpaired Student’s *t*-tests (**c**,**e**,**g**–**i**); *p*-value was calculated using paired Student’s *t*-test (**b**).

**Figure 3 cancers-14-04188-f003:**
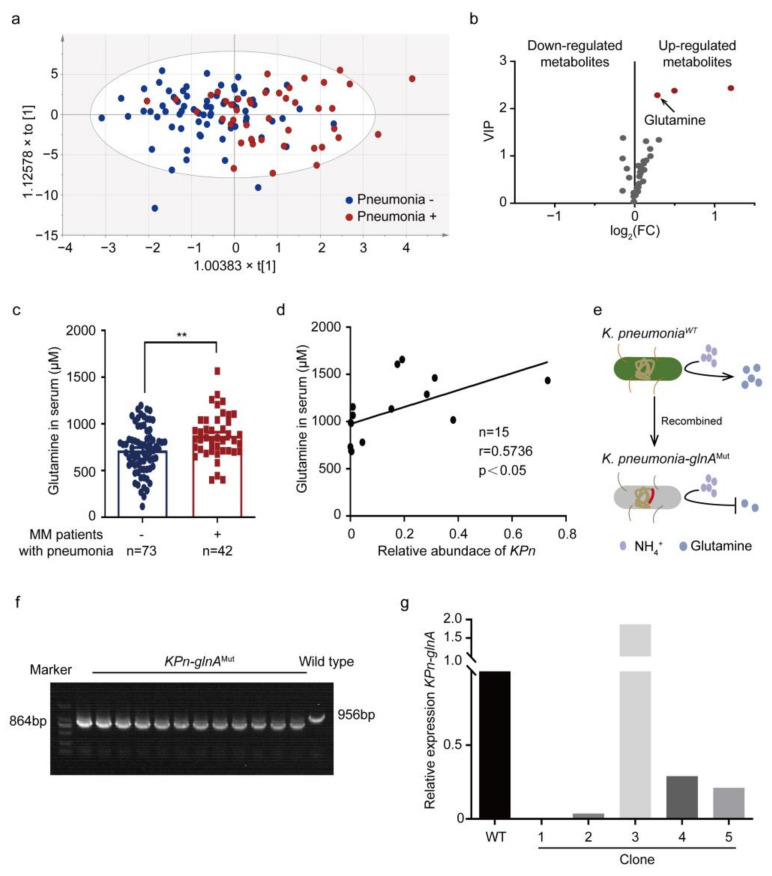
Glutamine is elevated in PB plasma of MM patients with pneumonia. (**a**) A plot of OPLS-DA scores for the PB plasma derived from MM patients without pneumonia (n = 73) and with pneumonia (n = 42); (**b**) differential metabolites based on VIP, FC, and *p*-value. Annotated metabolites were analyzed at the univariate level by using MetaboAnalyst 3.0 software with the nonparametric Student’s *t*-test; (**c**) targeted metabolomic assays of glutamine in the PB plasma derived from MM patients without pneumonia and with pneumonia (mean ± SEM); (**d**) the Pearson correlation between relative abundance of intestinal *K. pneumonia* and serum glutamine concentrations from paired fecal and serum samples in newly diagnosed MM patients; (**e**) *K. pneumonia* containing mutant *glnA* was constructed; (**f**) the DNA agarose gel electrophoresis of *glnA* from *K. pneumonia* wild type and mutant strain; (**g**) relative expression of intestinal *K. pneumonia*-*glnA* in *K. pneumonia* wild type and mutant type; ** *p* < 0.01; *p*-value was calculated using two-tailed unpaired Student’s *t*-tests (**c**); *p*-value was calculated using Pearson correlation analysis (**d**).

**Figure 4 cancers-14-04188-f004:**
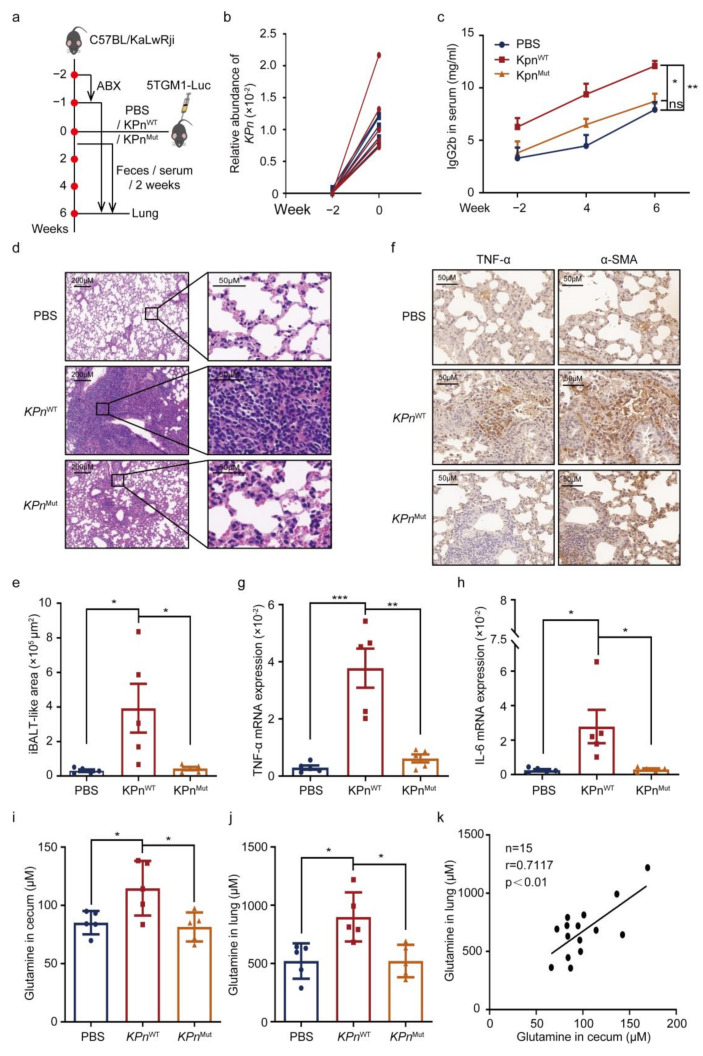
Intestinal *K. pneumonia* synthesizes glutamine to bolster pneumonia in MM in vivo. (**a**) Schematic of in vivo experimental workflow (three groups: control group (PBS), colonization of *K. pneumonia* with wild type (*KPn*^WT^), and mutant type containing mutant *glnA* (*KPn*^Mut^)); (**b**) relative abundance of intestinal *K. pneumonia* in the *KPn*^WT^ and *KPn*^Mut^ mice at week −2 and week 0 by qPCR; (**c**) serum IgG2b concentrations in PBS, *KPn*^WT^, and *KPn*^Mut^ groups by ELISA (mean ± SEM); (**d**) representative images of H&E staining of the lung from PBS, *KPn*^WT^, and *KPn*^Mut^ groups, respectively; (**e**) the iBALT-like area in PBS, *KPn*^WT^, and *KPn*^Mut^ groups (mean ± SEM); (**f**) representative images of immunohistochemistry staining of TNF-α and α-SMA in the lung from PBS, *KPn*^WT^, and *KPn*^Mut^ groups; (**g**,**h**) mRNA expression of TNF-α and IL-6 in the lung from PBS, *KPn*^WT^, and *KPn*^Mut^ groups by using qPCR (mean ± SEM); (**i**,**j**) concentrations of glutamine in cecum and lung of PBS, *KPn*^WT^, and *KPn*^Mut^ groups by HPLC (mean ± SEM); (**k**) the Pearson correlations between glutamine concentrations of cecum and lung in MM mice; * *p* < 0.05, ** *p* < 0.01, *** *p* < 0.001; *p*-values were calculated using two-tailed unpaired Student’s *t*-tests (**e**,**g**–**j**); *p*-value was calculated using paired Student’s *t*-test (**b**); *p*-value was calculated using one-way ANOVA (**c**); *p*-value was calculated using Pearson correlation analysis (**k**).

**Figure 5 cancers-14-04188-f005:**
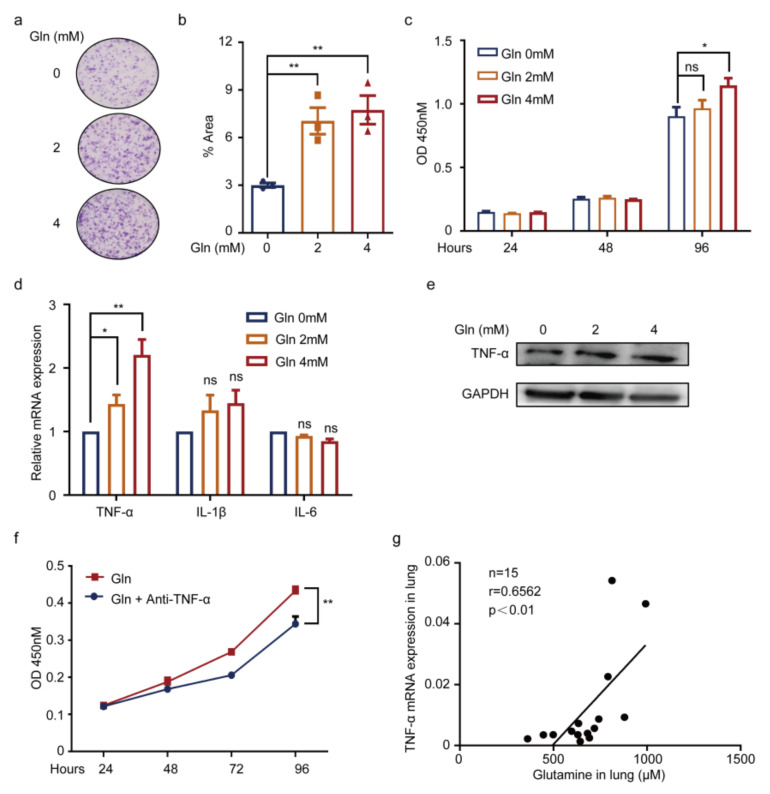
Glutamine contributes to pneumonia by promoting TNF-α expression of WI38 cells. (**a**,**b**) Images (**a**) and quantification (**b**) of plate clone of WI38 cells cultured in DMEM media with exogenous glutamine supplement (0 mM, 2 mM, 4 mM) (mean ± SEM, n = 3 for each group); (**c**) CCK-8 assay of WI38 cells cultured with additional glutamine supplement (0 mM, 2 mM, 4 mM) for 24 h, 48 h, and 96 h (mean ± SEM, n = 3 for each group); (**d**) mRNA expression of TNF-α, IL-1β, and IL-6 in WI38 cultured with exogenous glutamine supplement (0 mM, 2 mM, 4 mM) for 24 h; (**e**) Western blots of TNF-α and GAPDH in WI38 cells with additional glutamine supplement (0 mM, 2 mM, 4 mM) for 72 h; (**f**) WI38 cells cultured with additional glutamine (2 mM) combined treatment with or without anti-TNF-α by a CCK-8 assay (mean ± SEM, n = 3 for each group); (**g**) the Pearson correlations between lung glutamine concentrations and lung TNF-α mRNA expression in MM mice (n = 15); * *p* < 0.05, ** *p* < 0.01; *p*-values were calculated using two-tailed unpaired Student’s *t*-tests (**b**–**d**,**f**); *p*-value was calculated using Pearson correlation analysis (**g**).

**Figure 6 cancers-14-04188-f006:**
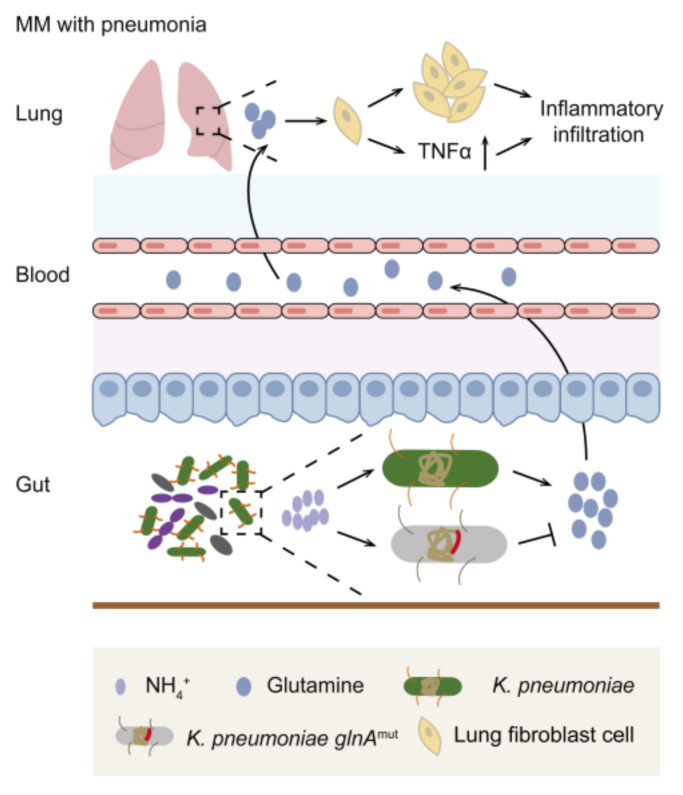
Schematic of working hypothesis.

**Table 1 cancers-14-04188-t001:** The correlation between intestinal *K. pneumonia* abundance and clinical characteristics of newly diagnosed MM patients.

Patients’ Characteristics	High *K. pneumonia* (n = 10; n/N × 100%)	Low *K. pneumonia* (n = 19; n/N × 100%)	*p*-Value
Sex			0.6999
Male	6/10 (60.00)	9/19 (47.37)	
Female	4/10 (40.00)	10/19 (52.63)	
Age (Year)	59.8 ± 9.19	59.53 ± 11.73	0.9495
Subtype of immunoglobulin			1
IgA	1/10 (10.00)	3/19 (15.79)	
IgG	6/10 (60.00)	12/19 (63.16)	
Subtype of immunoglobulin			0.2701
κ	6/10 (60.00)	7/19 (36.84)	
λ	4/10 (40.00)	12/19 (63.16)	
DS stage			0.6942
I + II	3/10 (30.00)	8/19 (42.11)	
III	7/10 (70.00)	11/19 (57.89)	
ISS stage			1
I	1/10 (10.00)	2/19 (10.53)	
II + III	9/10 (90.00)	17/19 (89.47)	
Pneumonia			0.0418 *
Yes	7/10 (70.00)	6/19 (31.58)	
No	2/10 (20.00)	13/19 (68.42)	
LDH (U/L)	197 ± 55.19	191.32 ± 43.2	0.7619
Immunoparesis			1
Yes	10/10 (100.00)	18/19 (94.74)	
No	0/10 (0)	1/19 (5.26)	
Plasma cells in bone marrow (%)	19.96 ± 16.11	30.36 ± 24.96	0.2713
Ca^+^ (mmol/L)	2.37 ± 0.48	2.18 ± 0.28	0.1792
Creatinine (μmol/L)	141.4 ± 105.4	111.02 ± 74.6	0.3747

* *p* < 0.05 The correlations of intestinal *K. pneumonia* abundance with clinical characteristics were measured using the two-tailed Fisher’s exact *t*-test and student’s *t* test.

**Table 2 cancers-14-04188-t002:** The correlation between glutamine concentration and clinical characteristics of newly diagnosed MM patients.

Patients’ Characteristics	High Glutamine (n = 57; n/N × 100%)	Low Glutamine (n = 58; n/N × 100%)	*p*-Value
Sex			0.2234
Male	33/57 (57.89)	27/58 (46.55)	
Female	24/57 (42.11)	31/58 (53.44)	
Age (Year)	59.7 ± 10.04	56.38 ± 10.84	
Subtype of immunoglobulin			0.4002
IgA	14/57 (24.56)	10/58 (17.24)	
IgG	29/57 (50.88)	33/58 (56.90)	
IgD	3/57 (5.26)	6/58 (10.34)	
Subtype of immunoglobulin			0.5803
κ	28/57 (49.12)	32/58 (55.17)	
λ	28/57 (49.12)	26/58 (44.83)	
DS stage			0.1136
I	5/57 (8.77)	1/58 (1.72)	
II + III	52/57 (91.23)	57/58 (98.28)	
ISS stage			0.0534
I	13/57 (22.81)	10/58 (17.24)	
II	19/57 (33.33)	10/58 (17.24)	
III	25/57 (43.86)	38/58 (65.52)	
pneumonia			0.0093 **
Yes	27/57 (47.36)	14/58 (24.14)	
No	30/57 (52.63)	44/58 (75.86)	
LDH(U/L)	158.25 ± 65.67	220.79 ± 239.49	0.0598
Immunoparesis			0.7051
Yes	50/57 (87.72)	53/58 (91.38)	
No	6/57 (10.53)	5/58 (8.62)	
Plasma cells in bone marrow (%)	31.73 ± 23.36	34.48 ± 27.34	0.6782
Ca^+^ (mmol/L)	2.37 ± 0.39	2.28 ± 0.31	0.1685
Creatinine (μmol/L)	115.14 ± 128.56	113.71 ± 92.67	0.9455

** *p* < 0.01 The correlations of glutamine expression with clinical characteristics were measured using the chi-square test and student’s *t* test.

## Data Availability

Metagenomic sequencing files for samples of newly diagnosed MM patients used in this study were deposited in the public database of the National Omics Data Encyclopedia (NODE) under project number OEP000194, with the available URL at https://www.biosino.org/node/review/detail/OEV000075?code=SEPGGE5F (accessed on 22 August 2022).

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
