# Peer review of "Intestinal Klebsiella pneumoniae Contributes to Pneumonia by Synthesizing Glutamine in Multiple Myeloma"

_cancers, 2022, doi:10.3390/cancers14174188_

Round 1

Reviewer 1 Report

Wang, et al. provided significant insight of intestinal K. pneumonia enrichment is associated with pneumonia in MM and cause pneumonia by synthesizing glutamines to promote TNF-α expression. Overall, manuscript is well written and can be improved further. I have few concerns:

·       Why the abundance of other commonly found gut bacteria was low (e.g., Lactobacillus) in MM without pneumonia patients (Figure 1b). Explain?

·       Patients sample size is very less (Figure 1b).

·       Authors should show 16s rRNA sequencing results after antibiotic treatment and pneumonia transplant (Figure 2). This will help to eliminate the possibilities of false results.

·       Did the patient’s pneumonia have diagnosed with any other disease or infection? This should be mentioned in the method section.

·       It is important to do all experiments in triplicates. Author needs to show original western blot results from each experiment.

·       Which mice was used for experiments, male or female?

·       What is the effect of diet the abundance of pneumonia?

·       It is important to know whether intestinal K. pneumonia is translocating form gut to lung?

Reviewer 2 Report

The article presents an interesting hypothesis for the role of intestinal K.pneumonia in pneumonia in newly diagnosed multiple myeloma patients through synthetization of glutamine which leads to elevated secretion of TNF-α in the lung. This is supported by samples from NDMM patients as well as in mouse models. The article is interesting, and the investigators use various methods to support this hypothesis. There are however a few things that need to be clarified.

Comments:

1.  Citation 4 is an article with a cohort with MGUS, not MM patients and should be omitted.

2.  Please clarify the following: “To figure out the incidence of pneumonia in NDMM, we retrospectively summarized 193 the cases of Xiangya Hospital in the past year.” Did you review all NDMM in the hospital in the past year (193 patients) and they had undergone 194 chest x-rays/CTs whereof 44% had new infiltrates as well as clinical symptoms of pneumonia? Is the 44% the incidence in all NDMM, or only in those with chest radiography?

3. Please add information on when the faecal samples were gathered – before or after treatment for the pneumonia? Could antibiotic treatment for the pneumonia have affected the results? If that is the case this should be addressed or added as a limitation in discussion.

4.  Do you have any cultures/information regarding what agent caused the pneumonia in the NDMM patients? Was it viral or bacterial? Did you rule out COVID-19?

5.  In results it is stated that intestinal K. pneumonia abundance correlated with the incidence of pneumonia. Could this correlation be explained by confounders such as antibiotic treatment or immunoparesis? Did you adjust for confounders? 

6.  If you have information on previous antibiotic exposure or immunoparesis this should be added to table 1 and 2.

7.  You hypothesize that K. pneumonia leads to pneumonia through elevated glutamine. Did you measure glutamine in the initial 29 patients where shotgun metagenomic sequencing of fecal samples was performed? In table 3 you present a figure with paired fecal and PB plasma samples in 15 NDMM patients – are these different patients? Did they have pneumonia at the time of sampling? Please clarify in results.

8. In 3.2 in results, you state that “The above-mentioned results described the more serious pneumonia in KPn mice, hinting that intestinal K. pneumonia accelerated pneumonia in 5TGM1 MM mice”. How was pneumonia confirmed in the mice? Or do you mean that the KPn mice had more inflammation in the lung tissue? Please describe more clearly.

9. In results, a section regarding strengths and limitations of the research is lacking. The discussion regarding treatments to balance the gut microbiota/probiotics could be shortened since this is not investigated in this study.

Reviewer 3 Report

This paper is interesting and can be accepted for publication.
